# A Comprehensive Overview of Baboon Phylogenetic History

**DOI:** 10.3390/genes14030614

**Published:** 2023-02-28

**Authors:** Gisela H. Kopp, Riashna Sithaldeen, Franziska Trede, Franziska Grathwol, Christian Roos, Dietmar Zinner

**Affiliations:** 1Zukunftskolleg, University of Konstanz, 78457 Konstanz, Germany; 2Department of Biology, University of Konstanz, 78457 Konstanz, Germany; 3Centre for the Advanced Study of Collective Behaviour, University of Konstanz, 78457 Konstanz, Germany; 4Department of Migration, Max Planck Institute of Animal Behavior, 78315 Radolfzell, Germany; 5Academic Development Programme, Centre for Higher Education and Development, University of Cape Town, Cape Town 7700, South Africa; 6Cognitive Ethology Laboratory, German Primate Center, Leibniz Institute for Primate Research, 37077 Göttingen, Germany; 7Gene Bank of Primates and Primate Genetics Laboratory, German Primate Center, Leibniz Institute for Primate Research, 37077 Göttingen, Germany; 8Department of Primate Cognition, Georg-August-University, 37073 Göttingen, Germany; 9Leibniz-ScienceCampus Primate Cognition, 37077 Göttingen, Germany

**Keywords:** mtDNA, hybridization, introgression, *Papio*, phylogeography

## Abstract

Baboons (genus *Papio*) are an intriguing study system to investigate complex evolutionary processes and the evolution of social systems. An increasing number of studies over the last 20 years has shown that considerable incongruences exist between phylogenies based on morphology, mitochondrial, and nuclear sequence data of modern baboons, and hybridization and introgression have been suggested as the main drivers of these patterns. Baboons, therefore, present an excellent opportunity to study these phenomena and their impact on speciation. Advances both in geographic and genomic coverage provide increasing details on the complexity of the phylogeography of baboons. Here, we compile the georeferenced genetic data of baboons and review the current knowledge on baboon phylogeny, discuss the evolutionary processes that may have shaped the patterns that we observe today, and propose future avenues for research.

## 1. Hybridization in Primates

Increasing evidence for gene flow among divergent lineages challenges the notion of evolution as a simple dichotomic branching process, in which one species diverges into two new ones that subsequently evolve independently [1]. Instead, past and ongoing gene flow between divergent taxa leads to network-like, or reticulate, phylogenetic relationships [2]. This natural hybridization is ubiquitous across the animal kingdom, varying in extent and magnitude [3,4]. The evolutionary consequences of hybridization are highly variable and include “reverse speciation” [5]; reinforcement of reproductive isolation [6]; the formation of stable hybrid zones [7,8]; hybrid speciation [4]; (adaptive) introgression, also from so-called “ghost” lineages [9,10,11]; and mitochondrial capture or nuclear swamping [12]. All of these effects of hybridization have been documented in primates in general, and in baboons (genus *Papio*) in particular, both in historic and extant populations [12,13,14,15,16,17,18].

There is increasing acknowledgment of the importance of admixture (via hybridization and introgression) as an evolutionary process. This can be directly attributed to the increasing use of genetic—and more recently, broad-scale genomic—data to reconstruct phylogenetic relationships. Incongruence between taxonomic classification schemes and phylogenetic trees, or between phylogenetic trees constructed from different datasets, e.g., nuclear and mitochondrial, are often the first indication for historical and/or ongoing hybridization. The shift from single-locus to multi-locus datasets has revealed discordances between different gene trees, which in some cases is evidence of hybridization events [19]. The term discordance is misleading in that it suggests conflicting evolutionary histories rather than different perspectives on one complex history that often involves reticulated evolution [20] and differences in gene flow between different parts of the genome [21,22]. For example, differences in gene flow arise in taxa with male dispersal and female philopatry, which is the predominant pattern in cercopithecines. Accordingly, the gene flow of female-specific markers, such as mitochondrial DNA (mtDNA), is geographically much more restricted than gene flow of biparentally or paternally inherited markers, leading to deeper divergence times for mitochondrial markers than for Y-chromosomal or autosomal markers in the same samples. In the case of this unidirectional gene flow crossing taxonomic borders, genomic introgression and nuclear swamping can occur [12]. This will become evident when comparing phylogenetic trees based on mtDNA and nuclear DNA (nDNA) [12]. Differential introgression can also arise when different genomic regions have different adaptive values. For example, genes that enhance the fitness of their carrier irrespective of potential environmental differences will flow more easily than genes for which the adaptive value is locally restricted or absent. This process has been documented for immune-related genes in a baboon hybrid zone [23]. One way to resolve discordance is to use population genomic approaches, which can reveal the mosaic structure of genomes and thus provide detailed data on ancestry and divergence patterns [17]. They can also help to elucidate complex histories of admixture and the introgression of neutral and adaptive alleles, respectively [24].

## 2. Disentangling the Phylogeny of Baboons

The taxonomy of baboons has been a matter of debate for decades [25,26,27]. Today, the pragmatic consensus is to follow a taxonomy with six phylogenetic species acknowledging Guinea baboons (*Papio papio*), olive baboons (*P. anubis*), hamadryas baboons *(P. hamadryas*), yellow baboons (*P. cynocephalus*), chacma baboons (*P. ursinus*), and Kinda baboons (*P. kindae*) [28]. Despite this general “working agreement” on species status, the phylogenetic relationships among these taxa have been difficult to disentangle. Behavioral and morphological characters, while valid to delimit taxa, appeared insufficient to reach a satisfactory understanding of the phylogenetic relationships among baboons, and the focus turned to the analysis of molecular data. Many of the conflicting results in phylogenetic reconstructions, initially considered as striking peculiarities (for example, the sometimes deep divergence of mitochondrial lineages within taxa and smaller mitochondrial distances between taxa), turned out to reveal the complex evolutionary history of baboons and similar observations in other genera have followed [13,14].

Newman et al. [29] and Wildmann et al. [30] described shared mitochondrial haplotypes in neighboring populations of hamadryas and olive baboons, hinting at the potential role of gene flow via hybridization in the evolutionary history of baboons. These findings led to intensive efforts to investigate the molecular phylogeny of baboons in greater detail, using samples of natural populations across the whole distribution range of the genus. Phylogenetic reconstructions based on mtDNA sequence data found several well-supported monophyletic clades, which fit the “north-south split” hypothesis (which includes the northern olive, Guinea, and hamadryas baboons as distinct from the southern yellow, Kinda, and chacma baboons [27,31,32]). However, these clades reflect the geographic origin of the analyzed samples rather than morphological classifications and reveal para- and polyphylies of the extant baboon taxa [32,33,34,35] (Figure 1 and Figure 2). Subsequent studies using complete mitochondrial genome sequences supported the fossil evidence of the southern African origin of the genus, the presence of a northern and a southern clade, and the para- and polyphyletic relationships between most baboon species [36,37]. Yellow baboons are represented in both the northern and the southern mitochondrial clade: northern yellow baboons cluster with eastern olive and hamadryas baboons, southern yellow baboons cluster with northern chacma baboons and form a sister clade to Kinda baboons. Yellow baboons from the Udzungwa Mountains in Tanzania even exhibit most likely a rare example of inverted intergeneric introgression. The yellow baboons from the Udzungwa Mountains carry mtDNA haplotypes closely related to that of the critically endangered and sympatric kipunji (*Rungwecebus kipunji*), while a different kipunji population has been most likely introgressed by yellow baboon mtDNA [16,37,38,39,40]. Olive baboons are also far from being mitochondrially monophyletic. Two deep branches in olive baboons are each more closely related to neighboring taxa: western olive baboons cluster with Guinea baboons, whereas eastern olive baboons cluster with hamadryas and northern yellow baboons. In sum, baboon mtDNA is sorted into geographical clusters of populations rather than recognized species, a pattern found also in other primate taxa (e.g., *Chlorocebus* spp. [41]).

Investigations based on nDNA markers, which are expected to better trace morphological variation, clearly identify genetic clusters that correspond to the taxonomic classification [18,42]. However, to date, nDNA studies have struggled to reach a satisfying geographic coverage of natural populations and may therefore represent incomplete pictures of the evolutionary histories. Species tree reconstructions based on nDNA are in general concordant with the evolutionary history derived from mtDNA analyses, but they differ in the detailed relationships among species within the northern and southern clade, respectively. Based mainly on samples from captive individuals without clear geographic provenance, Boissinot et al. [42] identified chacma baboons as diverging first and yellow baboons second, and a northern clade with hamadryas, olive, and Guinea baboons, in which Guinea baboons diverge first. Y-chromosomal data also support the “north-south split” hypothesis, with Kinda baboons diverging first in the southern clade and hamadryas baboons diverging first in the northern clade [43]. Phylogenetic reconstructions based on *Alu* insertion polymorphisms found intergeneric introgression between *Papio* and *Theropithecus*, a southern clade in which chacma and Kinda baboons are most closely related to the exclusion of yellow baboons, and a northern clade in which Guinea and olive baboons are most closely related to the exclusion of hamadryas baboons [44,45]. This topology was further supported by genome-scale studies on baboon phylogenetic relationships, which both confirmed the “north-south split” hypothesis and emphasized the role that hybridization played throughout the evolutionary history of baboons [18]. These studies also strengthen two reasons for past difficulties in resolving phylogenetic relationships among the taxa: (i) fast radiation in both the southern and northern clade within a short time frame, and (ii) considerable sex-biased gene flow among lineages [18].

## 3. The Cradle of Baboons: Introgressive Hybridization between Southern Baboons

The origin of modern baboons in southern Africa and the occurrence of deeply diverged mitochondrial clades in this region have led to many studies focusing on this area. The three baboon species distributed here—yellow, chacma, and Kinda baboons—are separated into seven major mitochondrial clades [35,49,50] with strikingly different geographic range sizes and some overlap (Figure 1). They are characterized by significant ancient admixture. The two main chacma clades (A: southern chacmas, B1: northern chacmas) cover the distribution of chacma baboons. Southern yellow baboons comprise three clades east and west of Lake Malawi (B2a: southern yellows, west of Lake Malawi, B2b: southern yellows, east of Lake Malawi, B3: Luangwa Valley yellows). Kinda baboons comprise one clade (C). The West Tanzanian Mahale clade (H) with an unclear taxonomic assignment is located at the three-taxon border of Kinda, yellow, and olive baboons. The range of the southern chacma clade A overlaps largely with the ranges of phenotypical Cape (*P. u. ursinus*) and Ruacana chacma baboons (*P. u. ruacana*) (Appendix A) and stretches south and west of the Kalahari Desert, covering large parts of the baboon distribution in South Africa, western Namibia, and probably Angola. Note, however, that there is little information available for Angolan chacma baboons, which might constitute a clade on their own. Further, hybridization between chacma and Kinda baboons in Angola has not been investigated. The northern chacma clade B1 is spatially overlapping in part with the range of Cape chacma but predominantly with the range of gray-footed chacma baboons (*P. u. griseipes*). Genomically, this is likely the result of male-biased introgression from chacma baboon ancestors into southern yellow baboon populations. It can be subdivided into an eastern chacma clade in the very east of South Africa and Eswatini, and a northern clade stretching from northern Namibia (where it overlaps considerably with the southern chacma clade) and Botswana, north of the Kalahari to Zimbabwe, central Mozambique and southern Zambia, where it comes into contact with Kinda and Luangwa Valley baboons. The divergence time of the two major chacma mtDNA clades has been dated to around the Early Pleistocene between 2 and 1.5 million years ago (mya) [49,50]. Aridification cycles during the Late Pleistocene and the resulting isolation of chacma baboon populations in refugia probably lead to lineage divergence still evident today [51]. The expansion and contraction of suitable baboon habitats have also been shown for the Last Glacial Maximum [52], possibly further contributing to complex patterns of secondary contact and gene flow between the divergent mitochondrial lineages. Importantly, chacma and yellow baboons are characterized by mitochondrial paraphyly, which can probably be attributed to introgressive hybridization and nuclear swamping due to the invasion of chacma baboon males into originally yellow baboon populations further north [33,49] (Figure 3). Detailed investigations of phenotypic characters in the putative current contact zone of southern yellow and northern chacma baboons in Gorongosa National Park are suggested as evidence for past and/or ongoing hybridization between these populations [53], but genomic data have not found any support for recent gene flow [43]. Possibly, Luangwa Valley baboons represent hybrids between yellow, Kinda, and chacma baboons, but there is no genetic evidence available yet.

## 4. The Two Main Clades Mingling: Past and Present Hybridization of Baboons in Eastern Africa

A detailed investigation of baboons in eastern Africa is of particular interest, as this region was supposedly extremely affected by climate fluctuations in the Pleistocene [54,55,56] and the resulting expansion and retraction of habitat suitable for baboons and other savannah species [34,57,58]. Geological events, such as rifting or volcanism, during the Pleistocene [59,60,61] could have further constituted temporary barriers. The deepest split in the baboon mitochondrial phylogeny, between the northern and the southern mitochondrial clades, is geographically localized in East Africa (Figure 1 and Figure 2). A fine-scale analysis of mtDNA enabled the precise localization of the boundary between northern and southern mitochondrial clades within the distribution of yellow baboons in central Tanzania from the coast to the eastern shore of Lake Tanganyika along the Ugalla-Malagarsi and the Ruaha-Rufiji rivers [35]. Interestingly, the Ruaha-Rufiji rivers seem to constitute a dispersal barrier also for other taxa, e.g., dwarf galagos [62] and subspecies of *Colobus angolensis* [63]. The contact zone of the current distributions of phenotypically well-differentiated olive and yellow baboons is also localized in East Africa but does not correspond to the boundary between northern and southern mtDNA clades. Both species occur in a zone of overlap where intermediate forms have been described and a well-established hybrid zone has been investigated in great detail [17,24,64]. In this hybrid zone, however, there is no clear relationship between specific mtDNA clades and phenotypes [32]. Yellow baboons from Zambia, Malawi, and southern Tanzania cluster with northern chacma and Kinda baboons in the southern clade, with a deep divergence of baboons from the Mahale Mountains in western Tanzania (clade H) at the species border of yellow, Kinda, and olive baboons. Yellow baboons north of the Ruaha-Rufiji rivers, as well as eastern olive baboons, fall within the northern clade (clade G1), closely related to the hamadryas baboon mitochondrial clade [35] (Figure 1). The deeper divergence of clades in the southern lineage in comparison to the northern lineage likely points to longer periods of isolation, while a high degree of gene flow and frequent introgression events appear to characterize the northern lineage. The expansion of yellow baboons northwards into probably a previous (proto)hamadryas range led to introgressive hybridization and nuclear swamping, resulting in the coastal (G4) and northern (G1) clades of yellow baboons [34]. The disjunct distribution of clade G4, which also comprises olive baboons from western Ethiopia, is either a relict of a formerly wider distribution of this clade or a result of incomplete geographic sampling. The expansion of olive baboons into both hamadryas and yellow baboon ranges—still observable until today in active hybrid zones in the Awash National Park, Ethiopia [65,66] and the Amboseli National Park, Kenya [64,67]—most likely resulted in nuclear swamping and the strong para- and polyphyletic relationships in the northern lineage. This is most evident in clades G1 and G4, which comprise both yellow and olive baboons, and clade G3, which comprises both olive and hamadryas baboons. It should be noted that the previously described northeastern olive clade G2 loses support with denser sampling and should be collapsed into clade G3 [48].

## 5. A Comprehensive Phylogeographic Scenario for Baboons

To develop a scenario of baboon phylogeographic history, we compiled georeferenced genetic data of modern baboons [18,35,37,48,50,51,68] and merged the phylogenetic reconstructions from these multiple datasets in a geographic context. The genus *Papio* is not well represented in the fossil record, but some key findings in southern and eastern Africa provide a crucial, additional context [69,70,71,72,73]. Like many other taxa, including the human lineage [74], baboons have been affected by climate changes, fluctuations and associated changes in habitats since the Pleistocene [52,57,58]. While extant baboons are considered ecological generalists and occur in a wide range of habitats from semi-deserts to humid forests [75], they are most strongly associated with savannah-like habitats [58]. Hyperarid areas without open water sources as well as dense rain forests are most likely unsuitable habitats and constitute dispersal barriers. Hence, expansions and contractions of savannahs and savannah-like habitats in the Pleistocene influenced dispersal and vicariance [76]. During drier periods, savannah habitats expanded and dispersal via opened savannah-routes (corridors) became possible, allowing for range expansions and gene flow in areas of secondary contact. In contrast, during wetter (or extremely dry) periods, forests (or deserts) expanded, leading to more or broader barriers. Simultaneously, favorable habitats shrank, leading to the isolation of formerly connected areas and the populations within, which supposedly led to the divergence and independent evolution of populations in isolated demes. Male-biased dispersal in the majority of baboon taxa (except Guinea and hamadryas baboons [77]) and resulting higher levels of gene flow in male-associated markers leads to shallower divergence times in Y-chromosomal as compared to autosomal or mitochondrial phylogenies.

As evidenced by both fossil and genetic data, the origin of baboons can be roughly placed in southern Africa between southern Tanzania and northern South Africa. We have no further knowledge about this ancestral modern baboon, but it has been hypothesized to resemble fossil *P. angusticeps* and Kinda baboons [70,78]. At about 2.5 mya, two independent mitochondrial lineages split from this ancestral baboon [37]. The more recent estimate of divergence of about 1.4 mya based on nDNA [18] hints at a longer persistence of male-biased gene flow. The southern lineage gave rise to yellow, chacma, and Kinda baboons. Probably facilitated by the contraction of an equatorial forest belt, which had constituted a barrier to dispersal, and the opening of a savannah corridor between the Tanzanian coast and the Congo basin during more arid periods, the expansion of suitable baboon habitat facilitated the dispersal into northern areas and the formation of the northern lineage [34,35,78]. The north–south split of the baboon lineage follows a general biogeographic pattern for African savannah taxa and can be found for instance in black-backed jackals (*Lupulella mesomelas*), bat-eared foxes (genus *Otocyon*), aardwolf (genus *Proteles*), and Oryx (genus *Oryx*) [79,80]. However, it is not clear whether there was one wave of baboon dispersal or several. Similarly, it is unclear which of the possible paths the northern movement followed (along the coast, i.e., east of Lake Malawi, between Lake Malawi and Lake Tanganyika, or even west of Lake Tanganyika). In a fast radiation, the northern lineage gave rise to the hamadryas baboon lineage expanding northeastwards, and the lineage that further split into olive and Guinea baboons westwards [18]. Fossils resembling modern baboons and dating to 600 to 150 thousand years ago (kya) have been described from Ethiopian sites [72], thus supporting this scenario. Guinea and hamadryas baboons exhibit the lowest level of mitochondrial nucleotide diversity among all investigated taxa [81], which is consistent with strong bottleneck effects or small founding populations and strong genetic drift related to low effective population sizes.

Bringing together mtDNA-based phylogenies with results from nuclear genomic analyses and the known geographic distribution of baboon phenotypes reveals a complex pattern that suggests that dynamic processes have shaped this phylogeographic scenario (Figure 3). Multiple instances of hybridization and introgression, most likely triggered by isolation–reconnection processes due to climatic and habitat oscillations, can provide a satisfying explanation for the observed discrepancies among datasets in general, but not in detail, since the degree of uncertainty for the temporal resolutions of the genetic and paleo-climate relationships is still high. Nuclear swamping resulting from male-biased dispersal into neighboring species constitutes a main process. Chacma baboons expanded from a southern population (nowadays Cape and Ruacana chacmas) into areas previously occupied by yellow baboons, giving rise to gray-footed chacmas (clade B1) [49,50]. Yellow baboons also expanded northwards into habitat presumably occupied by ancestors of the current northern lineage (clades G1 and G4) [35]. The expansion of olive baboons appears to have contributed significantly to the northern clade. Clade K presumably represents the original olive baboon clade, but nuclear genetic data are still lacking. Eastern olive baboons have expanded into the ranges of (proto)hamadryas and yellow baboon populations (clades J, G1, G2, G4), both historically but also recently and currently [17,24]. The situation in western Africa is the least well-investigated, but there is also evidence for introgressive hybridization between Guinea and western olive baboons [37].

## 6. Outstanding Questions and Future Research Directions

There are still considerable gaps in our understanding of baboon phylogeography. A main hurdle is insufficient geographic sampling in some crucial areas, especially in zones of known or suspected secondary contact and regions between the ranges of major mitochondrial clades. This is, for example, the case in West Africa, where the distributions of Guinea and olive baboons meet. The exact location of this contact zone is unclear [82] but recent evidence suggests that introgression also has occurred in this region [37]. The lack of continuous sampling is most severe in olive baboons, where crucial areas in the center of their distribution (between Cameroon and Ethiopia) have not been adequately covered. The sampling gaps between clades K, J, and G, and clades D and E especially prevent a thorough understanding of the relationships among the different olive baboon clades, and hence, the phylogeography of this species and its relationship to Guinea and hamadryas baboons. Another open question is why Guinea and hamadryas baboons evolved a multi-level social system, whereas the third northern species, olive baboons, retained or re-evolved the ancestral uni-level system. It is unclear whether secondary contact and gene flow with members of the southern lineage might have played a role in the olive baboon case. The investigation of Kinda baboons has until now focused on their southeastern distribution and the contact zone with gray-footed chacma baboons, leaving the rest of the distribution and the contact zone with Ruacana baboons in Angola unexplored. It is also unclear whether Kinda baboons have or had contact with yellow or olive baboons along the western shore of Lake Tanganyika. Filling these sampling gaps should be a main target in the coming years to achieve a true understanding of the genus-wide variation. Field excursions to collect samples from natural populations in these crucial regions might not always be possible or can bear risks due to political conflicts. Furthermore, non-invasively collected samples might not yield the appropriate quantity and quality of DNA for whole genome analyses, but approaches to obtain genomic data from fecal samples have recently been developed and promise major advancements in the near future [83,84,85]. In some of these regions, baboon populations might have become locally extinct due to an increase in human population and habitat loss in recent decades. Samples from museum collections can help to fill these gaps by allowing to analyze genetic diversity and evolutionary history based on baboon specimens with known provenance that have been collected in recent decades and centuries, employing advanced museum genomic techniques [37,86,87,88,89,90].

In addition to the geographical coverage, a more complete picture of genetic diversity needs to be achieved. The scenario for baboon phylogeography is based on evidence from pan-African mtDNA data and we have not yet reached a satisfactory coverage of nuclear data from natural populations. To extract nuclear and genome-scale data, high-quality and -quantity DNA is required, which is most commonly obtained from blood or tissue samples from captured animals. Hence, most nuclear and genome-scale data available today are either derived from samples from captive animals [18,42] or from single well-investigated natural populations [17,23,24,65,91]. Efforts to expand the whole-genome analysis to a broader coverage of populations will provide a more fine-scale picture of population structure and differential admixture, and can further capture important functional genetic variation. Pan-African genome-scale nuclear data are needed to confirm and refine the phylogeographic scenario of baboons, which can provide insights into the evolution of behavioral variation in this genus.

## Figures and Tables

**Figure 1 genes-14-00614-f001:**
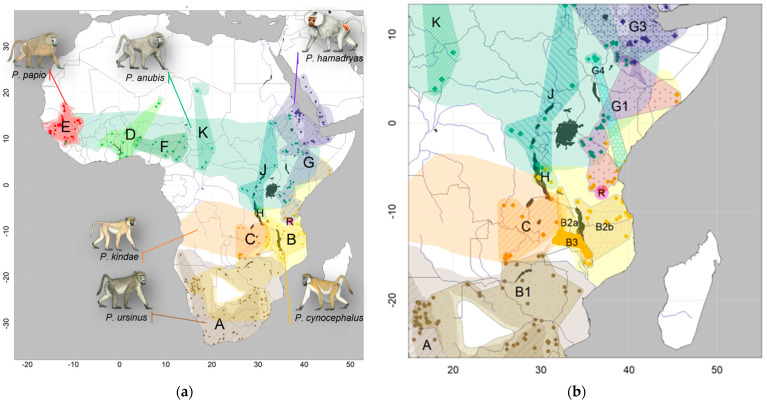
Geographic distributions of the six baboon species: (**a**) Overview of distribution ranges of the six baboon species according to IUCN (2020), colored by species (brown: *P. ursinus*, yellow: *P. cynocephalus*, orange: *P. kindae*, red: *P. papio*, green: *P. anubis*, purple: *P. hamadryas*). Main mitochondrial clade attributions are indicated by color-patterned regions and denoted with capital letters “A”-“K” and “R”. Circles, diamonds, and triangles represent the provenance of mtDNA markers, mitogenomes, and complete nuclear genomes, respectively, and are colored by species. (**b**) Close-up of the distribution of mitochondrial clades in the eastern distribution of baboons. Male baboon drawings by Stephen Nash, used with permission.

**Figure 2 genes-14-00614-f002:**
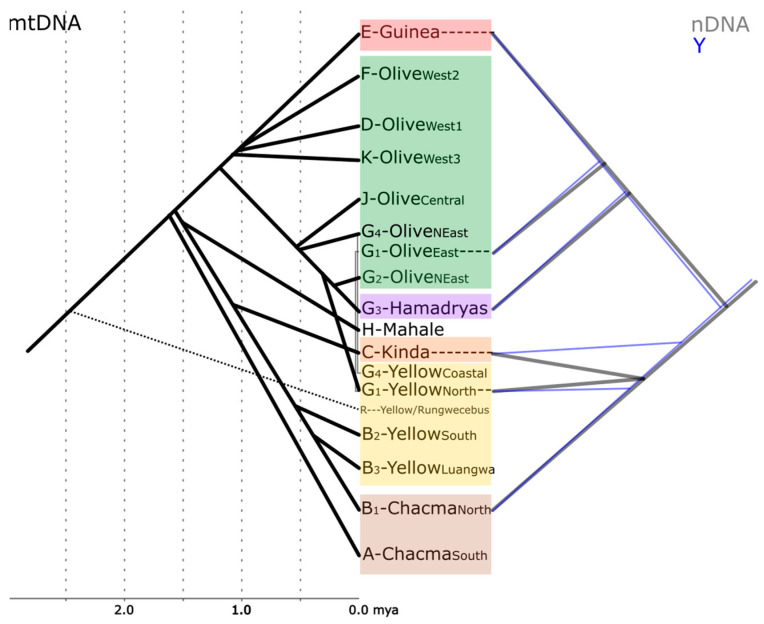
Sketch showing the phylogenetic relationships between baboons based on mtDNA [34,35,37,46,47,48], Y-chromosomes [43], and whole nuclear genomes [18]. Minor subclades are omitted for clarity. Divergence time estimates for Y-chromosomal data do not exist yet [43] and are inconclusive for nDNA [18] and are, therefore, not shown.

**Figure 3 genes-14-00614-f003:**
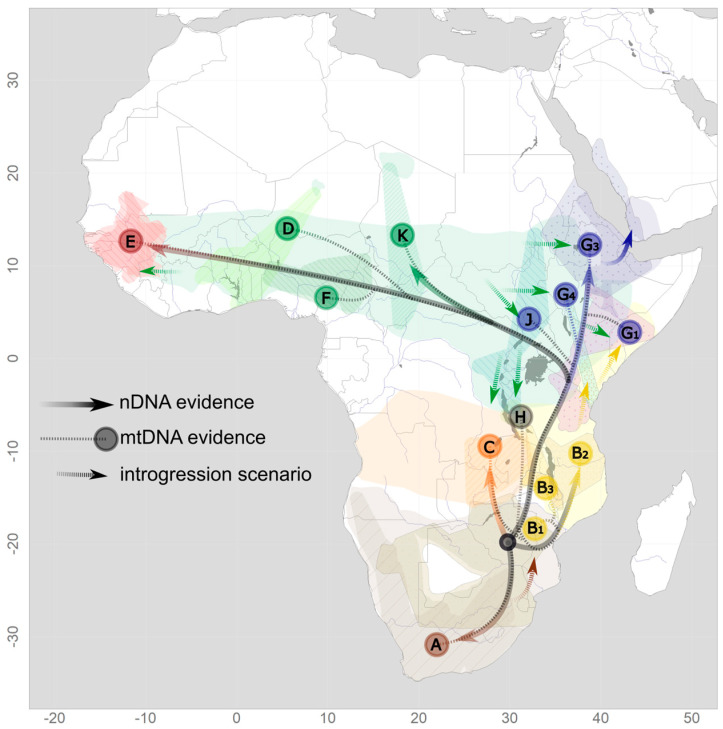
A phylogeographic scenario of baboons derived from nDNA (solid lines) and mtDNA (dotted lines) evidence. Letters indicate mtDNA clades as in Figure 2, with the color of the circle indicating the hypothesized proto-species.

## Data Availability

The data presented in this study and code to reproduce the results (Figure 1) are openly available on OSF via https://doi.org/10.17605/OSF.IO/76X4K.

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
