# Peer review of "A Comprehensive Overview of Baboon Phylogenetic History"

_genes, 2023, doi:10.3390/genes14030614_

Round 1

Reviewer 1 Report

This study describes the use of extant Baboons (six species within genus Papio) as a study system to investigate complex evolutionary processes, such as ongoing gene flow between divergent taxa (leading to network-like phylogenetic relationships) and evolutionary consequences of hybridization an introgression. The manuscript presents potentially interesting information, and in my opinion, it only needs minor revision. Here, I provide very few comments and observations.

Typos and minor:

-          Line 159. I think there is no exact correspondence between the mitochondrial clades and their description in the text. For example, within the figure B2 and B3 can be seen, but the text further refers to something slightly different (B2a or B2b). Moreover, the text between lines 158 and 162 is pretty unclear. It needs to be rewritten.

-          Lines 162-165. This can not be seen in the figure 1 since the distribution of Ruacana chacma baboons is not described within the figure.

-          Line 168. Remove “)” after “investigated”.

-           

Author Response

Thank you for your kind review. We adressed all your comments, please find our detailed responses to the specific items below.

Line 159. I think there is no exact correspondence between the mitochondrial clades and their description in the text. For example, within the figure B2 and B3 can be seen, but the text further refers to something slightly different (B2a or B2b). Moreover, the text between lines 158 and 162 is pretty unclear. It needs to be rewritten.

=> Thank you for pointing us to this. The division of clade B2 into two subclades is only depicted in the map in figure 1b. We did not specify minor subclades in the phylogenetic tree in figure 2 for simplicity. We included the sentence “Minor subclades are omitted for clarity” in lines 148/149 to clarify this. We rewrote lines 158 to 162 for better clarity.

Lines 162-165. This can not be seen in the figure 1 since the distribution of Ruacana chacma baboons is not described within the figure.

=> We added a supplementary map (Figure S1) with subspecies distributions to support this statement and added this to the text in line 165.

Line 168. Remove “)” after “investigated”.

=> Done

Reviewer 2 Report

This paper provides a comprehensive review of the current state of knowledge on the phylogeny of baboons (genus Papio) and their complex evolutionary processes. The authors highlight the significant incongruences observed between morphological, mitochondrial, and nuclear sequence data and suggest that hybridization and introgression are key drivers of these patterns. They also discuss the increasing details provided by advances in geographic and genomic coverage and propose future avenues for research in this field. Overall, this review is a valuable resource for anyone interested in understanding the evolutionary history of baboons and the impact of hybridization and introgression on speciation.

I support that the paper should be accepted. 

Author Response

Thank you for this kind review. We made minor changes to the manuscript to improve the phrasing and clarity.